# Multi-Objective Routing Optimization for 6G Communication Networks Using a Quantum Approximate Optimization Algorithm

**DOI:** 10.3390/s22197570

**Published:** 2022-10-06

**Authors:** Helen Urgelles, Pablo Picazo-Martinez, David Garcia-Roger, Jose F. Monserrat

**Affiliations:** iTEAM Research Institute, Universitat Politècnica de València, 46022 València, Spain

**Keywords:** multi-objective, quantum computing, quantum optimization algorithms, quantum routing optimization, 6G communication networks

## Abstract

Sixth-generation wireless (6G) technology has been focused on in the wireless research community. Global coverage, massive spectrum usage, complex new applications, and strong security are among the new paradigms introduced by 6G. However, realizing such features may require computation capabilities transcending those of present (classical) computers. Large technology companies are already exploring quantum computers, which could be adopted as potential technological enablers for 6G. This is a promising avenue to explore because quantum computers exploit the properties of quantum states to perform certain computations significantly faster than classical computers. This paper focuses on routing optimization in wireless mesh networks using quantum computers, explicitly applying the quantum approximate optimization algorithm (QAOA). Single-objective and multi-objective examples are presented as robust candidates for the application of quantum machine learning. Moreover, a discussion about quantum supremacy estimation for this problem is provided.

## 1. Introduction

Quantum computing has boosted worldwide interest in different research areas, including telecommunications. The advent of complex sixth-generation (6G) technologies suggests that a quantum computing approach may better serve some use cases for high-performance computing in wireless communications. Optimization in communication networks has been a hot research topic throughout the years. The evolution of telecommunications has led to thousands of devices being connected, including nodes of the Internet of Things (IoT), autonomous vehicles, user devices, sensors, etc. Routing optimization plays an important role in guiding data packets between network nodes to follow the best end-to-end paths (from the source to the destination) according to certain network conditions. Regardless of the type of network, a poor routing strategy may prove harmful to the overall performance of the network. Routing strategies involve the definition of a set of one or more paths over which communication between end devices takes place over a network.

Typically, in conventional multi-hop networks, only one of the desired objectives is optimized, whereas other objectives are assumed to be constraints of the problem. Multi-objective algorithms have been previously explored in the literature, with the discussion focused on comparing complexities and rates of convergence with respect to the number of network nodes. An example is [1], where the authors proposed two approaches: (i) an algorithm based on the non-dominated sorting-based genetic algorithm-II (NSGA-II), and (ii) a multi-objective differential evolution (MODE) algorithm. Many single-objective optimization techniques have also been proposed; even though they are in their majority (based on classical solutions), some remarkable examples of quantum approaches exist. In [2], the authors proposed the non-dominated quantum iterative optimization (NDQIO) algorithm, which exploits the parallelism property in quantum mechanics for finding the optimum of a multi-objective routing problem in wireless multi-hop networks. Another algorithm, the tree-based quantum algorithm (TQA) in [3], solves the delay-constraint multi-cast tree problem. The most significant advantage of TQA is that the solving speed is much faster and more noticeable when the network topology scale becomes more considerable.

Furthermore, inspired by the IoT concept, with millions of interconnected wireless devices acquiring data ubiquitously, ref. [4] presented a novel quantum computing-inspired (IoT-QciO) optimization technique for wireless networks. The approach is based on the maximization of data accuracy (DA) in a real-time environment of IoT applications. In another recent paper [5], a quantum particle swarm optimization (PSO) algorithm is proposed, which outperformed existing optimization algorithms in terms of precision and convergence speed for smart IoT parking applications. The same authors applied an enhanced version of the PSO to routing in an industrial IoT (IIoT) scenario [6]. Formulations of routing problems on quantum computers appear in [7], where the authors focused on vehicle routing problems with time window(s) (VRPTW) and investigated and compared the VRPTW from a quantum computing perspective. Additionally, in [8,9], the quantum approximate optimization algorithm (QAOA) was tested with good results for the vehicle routing problem (VHP), a generalization of the traveling salesman problem (TSP). Finally, quantum optimization for 6G wireless communication is addressed through two use cases, MIMO detection and LDPC decoding in [10]. The case studies focus on quantum annealing (QA) technology, but the gate model processors are also described as potential quantum computations for communication networks.

Motivated by 6G connectivity requirements, the possibilities offered by the quantum computation, and the crucial role of routing strategies in wireless communications, we present a multi-objective routing optimization use case using QAOA and quantum systems. We integrated the parameterized lexicographic heuristic method to solve routing problems with quantum computing as a novel approach in the literature, and the results are provided to argue quantum supremacy.

In this context, the contributions of the present paper are as follows:We solved single and multi-objective network routing problems in a 6G network scenario, formulating the first QAOA [11] generalization to multi-objective optimization problems applied to vehicular ad hoc networks (VANETs) to the best of the authors’ knowledge. As these problems are known to be Non-deterministic polynomial-time hardness (NP-hard) [12], quantum algorithms may help speed up the problem-solving process when the problem’s size makes classical algorithms struggle to find optimal solutions in a reasonable time.We presented a single-objective routing optimization, providing the steps for the problem solution through quadratic unconstrained binary optimization (QUBO) and the Ising model to create the Hamiltonian problem;We discuss the performance of QAOA concerning the number of layers.We proposed a multi-objective routing optimization based on the parameterized lexicographic heuristic method and QAOA;We conclude quantum supremacy expectations from insights on estimations collected from the literature review.

The rest of the paper is structured as follows: Section 2 exposes firstly a brief overview of Quantum Optimization Algorithms, focused on QAOA. In Section 2.2, an example of a single-objective routing problem executed on IBM quantum experience is presented. After that, Section 2.3 provides a multi-objective routing problem using QAOA and parameterized lexicographic method. Finally, in Section 3, a discussion about supremacy expectations is presented. The main conclusions are drawn in Section 4.

## 2. Quantum Routing Optimization

In most cases, when network operators or service providers design and control their networks in real-world settings, they first formulate an optimization problem corresponding to the desired communication network with the required parameters and then solve the problem using a computer. These problems are mainly integer optimization problems whose complexities require high computational resources. To solve these problems on classic computers, the GNU linear programming kit (GLPK) package can be used (as was the approach of [13]). This kit is intended to solve linear programming (LP), integer LP, and mixed-integer LP programming. Since the main result of the decision-making logic of a routing process is the recommended path to be followed, an integer variable may be assigned to each connection between each connected node. If the variable takes the value of 1, which means the path is taken. In other cases, the variable will be 0 if that way is not optimal to reach the desired objective. Primary objectives could involve minimizing battery costs for remote nodes, maximizing the throughput for each involved node, or minimizing the number of jumps (which have advantageous impacts on latency). When the number of nodes increases, the LP problem can become so big that classic computers struggle to find the optimal route. This happens because the number of combinations between nodes is so significant that the variable numbers scale rapidly, meaning that high computational resources are needed to land on a solution. This will be true for future 6G communication networks, which are expected to provide global coverage and space–air–ground–sea [14]. Considering the multiple requirements that 6G will need to address simultaneously, quantum computers and quantum algorithms could play even more significant roles in such optimization problems.

### 2.1. Quantum Optimization Algorithms

Quantum computation takes advantage of the properties of quantum mechanics to tackle optimization problems radically differently. Ideally, due to the superposition of quantum states, quantum computers can process all the data simultaneously to find the solution that optimizes the objective function. In contrast, present-day “near-term” quantum computing or “noisy intermediate-scale quantum” (NISQ) technology implement at most 50–100 qubits, and while they might be able to perform tasks that exceed the capabilities of classical computers, they also exhibit noise-related inaccuracies that complicate the demonstration of the advantages of practical quantum computers and limit the sizes of quantum circuits [15]. One of the goals in the NISQ era is to extract the maximum quantum computational power from current devices while developing techniques that will suit the “long-term” goal of fault-tolerant quantum computations. Consequently, new classes of algorithms have been developed for this kind of system. Most of the current NISQ algorithms are based on a hybrid quantum-classic arrangement, such as the variational quantum eigensolver (VQE) and QAOA [16].

VQE was introduced in 2014 [17] for chemistry applications and quantum mechanics to estimate the ground state energy of a molecule using shallow depth circuits. The ground state of energy is equivalent to finding the minimum eigenvalue and/or eigenvector of a matrix (Hamiltonian), which characterizes the molecule. Apart from being applicable in these fields, it has spread up its functionality to optimization problems; one can also use the VQE for optimizing a cost function by encoding it as a matrix whose ground state (minimum eigenvector) corresponds to the optimal solution of the problem. This idea also lies at the heart of QAOA.

Since these algorithms require a smaller circuit (a few quantum gates), it better preserves the coherent evolution of the system, allowing a higher probability of successful results, also beneficial for the available systems with just a few noisy qubits.

QAOA is a variational quantum algorithm (quantum–classical hybrid algorithm) due to its implementation through quantum circuits that depend on a set of variational parameters (β, γ). It was introduced by Farhi and Goldstone in 2014 [11] to solve the problem of finding out a cut whose size was at least the size of any other cut (MaxCut) on a regular graph. This algorithm is characterized by a lower bound for the ratio between the result obtained by the algorithm and the optimal cost (the “approximation ratio”) and depends on an integer p(layers)≥1. The quality of the approximation improves as p is increased, and the depth of the quantum circuit grows linearly p times the number of constraints. In fact, in [11], QAOA always found a cut that was at least 0.6924 times the size of the optimal cut.

QAOA uses a unitary U(β, γ) characterized by the parameters (β, γ) to prepare a quantum state |ψ(β,γ)〉. The goal of the algorithm is to find optimal parameters ($\beta_{opt}$, $\gamma_{opt}$), such that the quantum state $|\psi (\beta_{opt},\gamma_{opt})\rangle$ encodes the optimal solution to the problem [18].

To summarize, QAOA’s principle is to extract (measure) the quantum solution prepared by a quantum state in a variational quantum circuit. Then, a classical optimizer is used to tune the circuit parameters and minimize the measured expectation value. Figure 1 graphically represents this principle of operation.

### 2.2. Single-Objective Quantum Routing Optimization

Since communication networks consist of nodes and links; one of the main objectives is to find the minimum cost (in terms of battery consumption) to transmit the traffic from an origin node to a destination node. A network is represented by a graph G(V,E), where *V* is the set of vertices (nodes), *E* is the set of links (weights), and the link from node *i* to node *j* is expressed as (i,j)∈E. Figure 2 shows an example of a network. If node 1 is the source node and node 4 is the destination node, then the problem consists of finding the shortest path from node 1 to node 4 depending on the requirements, constraints, and/or objectives.

Generally, this kind of problem can be formulated as a cost (objective) function (Equation 1), which is minimized or maximized according to constraints (Equation 2) and (Equation 3) and variables’ bound (Equation 4) since they involve seeking the best configuration among a set of parameters to achieve the desired objectives. In this example, the cost values are not representative of a real scenario and take values from 1 to 10, implying a higher value and bigger battery cost. This simplification is made because the goal is not to accurately define a cost function but to test QAOA and elaborate supremacy predictions on routing problems. The following equations can be found in [13].

Objective:(1)$$\min/\max\left(\sum_{(i,j)\in E}C_{ij}X_{ij}\right)$$

Subject to:(2)$$\sum_{j:(i,j)\in E}X_{ij}-\sum_{j:(i,j)\in E}X_{ji}=1,if\;i=p,$$
(3)$$\sum_{j:(i,j)\in E}X_{ij}-\sum_{j:(i,j)\in E}X_{ji}=0,\forall_{i}\neq p,q\in V,$$
(4)$$0\leq X_{ij}\leq 1,\forall_{\left(i,j\right)}\in E.$$

For this example, the problem formulation is presented below:(5)$$\min\;(5X_{12}+8X_{13}+2X_{23}+7X_{24}+4X_{34})$$

Subject to:(6)$$X_{12}+X_{13}=1,$$
(7)$$X_{12}-X_{23}-X_{24}=0,$$
(8)$$X_{13}+X_{23}-X_{34}=0,$$

One common model that is suitable for solving combinatorial optimization problems in quantum computers is the quadratic unconstrained binary optimization, or QUBO for short. QUBO can embrace many models in combinatorial optimization; QUBO models were shown to be equivalent to the Ising model, which plays a crucial role in physics and particle interactions [19].

A formal definition of the QUBO model is given by:(9)$$\min/\max\;(X^{T}QX+C^{T}X+c)$$
where X is a vector of binary decision variables, *Q* is a square matrix of quadratic coefficients, and C is a vector of linear coefficients.

Before solving our problem with QAOA, it should be cast in QUBO form. Although our problem includes additional constraints, it can be effectively reformulated as a QUBO model by introducing quadratic penalties (*P*) into the objective function (Equation 10).
(10)$$\begin{array}{c} \min\;(5X_{12}+8X_{13}+2X_{23}+7X_{24}+4X_{34}\\ +P{\left(X_{12}+X_{13}-1\right)}^{2}\\ +P{\left(X_{12}-X_{23}-X_{24}\right)}^{2}\\ +P{\left(X_{13}+X_{23}-X_{34}\right)}^{2}) \end{array}$$

Arbitrarily choosing *P* to be equal to 27, the *Q* matrix and C vector are given by:(11)$$Q=\left[\begin{array}{ccccc} 54 & 27 & -27 & -27 & 0\\ 27 & 54 & 27 & 0 & -27\\ -27 & 27 & 54 & 27 & -27\\ -27 & 0 & 27 & 27 & 0\\ 0 & -27 & -27 & 0 & 27 \end{array}\right]$$
(12)$$C^{T}=\left[\begin{array}{ccccc} -49 & -46 & 2 & 7 & 4 \end{array}\right]$$

As mentioned before, the cost function can be mapped to a Hamiltonian in order to find the ground state energy of the system that is equivalent to the optimal solution. QAOA is defined by the problem Hamiltonian ($H_{P}$) (Equation 13), which contains the cost function, and the mixer Hamiltonian ($H_{M}$) [18], defined as the sum of single Pauli *X*-operators on all qubits (Equation 14).
(13)$$H_{P}|x\rangle =\left(x^{T}Qx+c^{T}x\right)|x\rangle =\left(\sum_{i,j=1}^{n}x_{i}Q_{ij}x_{j}+\sum_{i=1}^{n}c_{i}x_{i}\right)|x\rangle$$
(14)$$H_{M}=\sum_{i=1}^{n}X_{i}$$

To define the $H_{P}$ by Pauli *Z*-operators, the objective function should be formulated as the Ising spin model: $x_{i}=\frac{1-Z_{i}}{2}$
(15)$$\begin{array}{c} H_{P}=11\left(IIIIZ_{0}\right)-17.5\left(IIIZ_{1}I\right)-28\left(IIZ_{2}II\right)-17\left(IZ_{3}III\right)+11.5\left(Z_{4}IIII\right)\\ +13.5(IIIZ_{1}Z_{0}-IIZ_{2}IZ_{0}-IZ_{3}IIZ_{0}+IIZ_{2}Z_{1}I-Z_{4}IIZ_{1}I+IZ_{3}Z_{2}II-Z_{4}IZ_{2}II) \end{array}$$

The small, single-objective, four-node example (Figure 2) is represented by its corresponding variational quantum circuit based on $H_{P}$ and $H_{M}$ ($H_{P}+H_{M}$) in Figure 3. The initial prepared state is the equal superposition state through Hadamard (H) gates. The iterations required to reach the optimal results depend on the quantum system used. It was tested on “ibm_perth”, a seven-qubit IBM Quantum System, and 59 iterations were necessary using COBYLA as a classical optimizer to find $\beta_{opt}=0.28517317$, and $\gamma_{opt}=-5.05969577$.

It is quite simple to note that the optimum path for minimizing costs would be 1–2–3–4, resulting in 11 (5+2+4). Figure 4 shows the probabilities results according to the possible paths and β and γ values. As expected, the |10101〉 state has the higher probability that corresponds with the $X_{12},X_{13},X_{23},X_{24},X_{34}$ where $X_{12}=1,X_{23}=1,X_{34}=1$.

While this small example may seem simple (this specific case took five qubits) when the number of nodes increases, the problem becomes classically intractable. Note that while a bigger network could not be solved on available quantum computer hardware because of the number of qubits required; this does not mean that larger problems cannot be solved. It depends on the number of available qubits. Furthermore, it must be mentioned that despite the QAOA result being 11, identical to the classical solution, in more complex problems by its nature, QAOA might provide only a good approximate near-optimum solution.

Additionally, to test how the algorithm performs according to the number of QAOA layers, simulations with 500 shots were executed and p=1,2,3. The theoretical accuracy provided by QAOA improves with higher values of p as was already anticipated in Section 2.1. Figure 5 illustrates how the probability of obtaining the right solution increases with higher values of p. However, it turns out that for implementation on a real device, this improvement is not remarkable because the depth of the quantum circuit has a negative impact on the noise level.

Furthermore, Algorithm 1 below, shows the pseudocode of the algorithm proposed for the routing when following the steps to programming the single target routing problem.
**Algorithm 1:** Routing Optimization using QAOA.1:**from** pyqubo **import**
Array,Constraint,Placeholder  Design the network graph:2:edges=[(1,2),(1,3),(2,3),(2,4),(3,4)], weights=[5,8,2,7,4]3:x=Array.create(shape=len(edges))4:**for**iteration=1,2,…**in** range(len(edges)) **do**5:    $f_{cost}+=$Constraint(weights[i]*x[i])6:**end for**7:$f_{cost}+=p*$Constraint$\left({\left(x\left[0\right]+x\left[1\right]-1\right)}^{2}\right)$8:$f_{cost}+=p*$Constraint$\left({\left(x\left[0\right]-x\left[2\right]-x\left[3\right]\right)}^{2}\right)$9:$f_{cost}+=p*$Constraint$\left({\left(x\left[1\right]+x\left[2\right]-x\left[4\right]\right)}^{2}\right)$  Create the problem Hamiltonian and mixer Hamiltonian: $x_{i}=\frac{1-Z_{i}}{2}$
(16)$$H_{P}+H_{M}$$  Create the QAOA circuit according to the linear and quadratic coefficients of $H_{P}$:10:linear_coefficients(lc)=[11.0,−17.5,−28.0,−17.0,11.5]11:quadratic_coefficients(qc)={(0,1):13.5,(0,2):−13.5,(0,3):−13.5,(1,2):13.5,(1,4):−13.5,(2,3):13.5,(2,4):−13.5}12:Circuit (num_qubits,param,n_layers,lc,jc):13:circ = QuantumCircuit(num_qubits)  Initial state (H gates):14:**for** qubit **in** range(circ.num_qubits) **do**15:    circ.h(qubit)16:**end for**  Problem Hamiltonian:17:**for** qubit **in** range(circ.num_qubits)) **do**18:    circ.rz(lc[qubit]*param,qubit)19:**end for**20:**for** key **in** qc.keys()) **do**21:    circ.rzz(qc[key]*param,key[0],key[1])22:**end for**  Mixer Hamiltonian:23:**for** qubit **in** range(circ.num_qubits) **do**24:    circ.rx(param,qubit)25:**end for**26:circ.measure(range(num_qubits), range(num_qubits))  27:Compute the expectation values according to the measurement results.  28:Optimize classically to find β and γ, with scipy.optimize.  29:Repeat the process until $\beta_{opt}$ and $\gamma_{opt}$ optimum are found.

### 2.3. Multi-Objective Quantum Routing Optimization

When different performance metrics need to be considered in communication networks, the optimization problem becomes multi-objective. In this regard, those metrics could be to reduce the cost to provide a better quality of service, improve the throughput, or minimize the number of hops taking care of the network’s latency. If only one parameter is considered in the link, the other parameters will probably not follow the requirements for a determined service. As a result, multi-objective problems are purposed to obtain solutions satisfying the multiple criteria for each scenario.

Solving multi-objective problems does require more computation power than single-objective problems. In addition, multi-objective problems do not have global optima. If one objective is optimized, it is likely that the others will not be. Equilibrium needs to be found to cover all the requirements desired; each equilibrium point is known as a Pareto optimal point.

From heuristics, there are multiple ways to find Pareto’s solutions. The parameterized lexicographic method is presented in this paper. This method needs to order objectives by importance. The optimization will have as many stages as objectives formulated. The first stage will solve the first objective without including any other one. The second stage will solve the second objective, including a margin constraint inherited from the first stage. The third stage will include inherited constraints from the first and second and so on. For example, if minimizing the battery cost is the most important objective, the cost result obtained from the first stage will have a deviation from the optimum allowed in the second stage. This deviation is parameterized using slack parameters α. If an objective wants to be minimized, the slack parameter of the inherited constraint, $\alpha_{n}$ will be higher than 1 (allowing its increase from the optimal) and if maximized lower than 1, (allowing its decrease from the optimal).

This subsection presents a practical case of multi-objective routing optimization using QAOA and the lexicographic method. Note that before solving our problem with QAOA, it should be cast in QUBO form as outlined previously in Section 2.2; the same steps presented in the pseudocode were followed, not included here for brevity. This case included six nodes following the scheme shown in Figure 6. Three parameters were optimized: i) the battery cost, ii) the available throughput for the served nodes, and iii) the number of hops from the origin to the destiny. The battery cost and the number of hops were minimized, and the throughput maximized. Since the number of qubits available was not enough, the problem was simulated on IBM Quantum first. In addition, it was executed on IBM Cloud through Qiskit Runtime [20] available on IBM Quantum systems and IBM Cloud. Qiskit Runtime is an architecture and programming service offered by IBM that allows users to optimize workloads and efficiently execute them on quantum systems. Runtime works based on programming via primitives such as Sampler, Estimator, and in our case, QAOA. The system used was imb_algiers, which has 27 qubits.

The first objective of the lexicographic approach was to minimize the battery cost, according to (Equation 1). All the constraints (Equation 2), (Equation 3), and (Equation 4), shown in Section 2.2 were added to the model. Following the same procedure, the solution gave the path 1–3–5–6, offering services to four nodes, which meant three hops. The total battery cost was 4, and the throughput obtained was 4.

Once the battery cost was optimized, the solution found was added as an inequity constraint with the alpha parameter, giving some clearance to the cost while optimizing the throughput. This parameter in this example was set to $\alpha_{c}=2$. Equation (17) was the added constraint to the model in this second step
(17)$$C_{ij}X_{ij}\leq \left(C_{ij}X_{ij}^{*}\right)\alpha_{c}.$$

The next objective was the throughput, calculated as the sum of the throughput values for all of the nodes served. The objective is shown in Equation (Equation 18). As was done with cost, a simplification was also made to the calculus of the throughput values for each path, not being the ones used representative of a real scenario. In this case, higher values are better since this objective wants to be maximized.
(18)$$\max\sum_{(i,j)\in E}Thr_{ij}X_{ij}.$$

The results changed, increasing the throughput to 10 and the cost to 7. In addition, the number of hops increased to 4. The path taken was 1–3–4–5–6. This means a sacrifice for both cost and hops to increase the throughput. Since multi-objective optimization does not have a correct result, this result is as valid as the one obtained in the first stage, and the final decision is made by the decision-maker.

As was done with the cost, an alpha parameter was added to the model, giving clearance to the throughput in the minimization of the last objective, the hops. This parameter was fixed to $\alpha_{thr}=0.4$. The reason for this parameter being that small was due to the tiny size of the problem, which required big clearances to change from one solution to another. This means that Equation (Equation 19) needed to be included in the model. Equation (17) was also added to the constraints for the last step of the multi-objective optimization process, the minimization of the number of hops.
(19)$$Thr_{ij}X_{ij}\geq \left(Thr_{ij}X_{ij}^{*}\right)\alpha_{Thr}.$$

The last objective, the number of hops, was minimized. The objective is shown in Equation (Equation 20).
(20)$$min\sum_{(i,j)\in E}H_{ij}X_{ij}.$$

All constraints in (Equation 2), (Equation 3), (Equation 4), (17), and (Equation 19) were added to the model.

The result obtained for the hops was 3, the cost was also reduced to 4, but throughput decreased to 4. This gave the same result as the one obtained in the first stage of the optimization, following the path 1–3–5–6.

The first step of the lexicographic method only considered minimizing the cost. As a result, even though only three hops were performed, throughput was not really good. When maximizing throughput, the cost was slightly increased and so were the hops. Finally, for this particular case, the solution taking into account the three objectives was equivalent to the solution that we would obtain from only considering the cost objective. The result-deriving process was forced to showcase the operation of the method. However, other combinations of alpha parameters or a change in the order of the objectives would change the results. Problems of a bigger size, which unfortunately could neither be run on a QC nor simulated, will be more dependent on changes of alpha, altering the final result upon small variations of alpha. The final results obtained from this method can be summarized in Table 1.

## 3. QAOA Supremacy Expectations

QAOA is one of the most promising candidates for achieving quantum supremacy in optimization problems, which means outperforming the best-known classical algorithms on a given problem. This algorithm can solve scheduling, data analysis, and machine learning optimization problems. This paper used it to solve a hard integer routing optimization problem with good results. However, a higher number of qubits with relatively low noise is necessary to outperform the classic computation solution.

In the last few years, quantum computers have entered a new phase, where qubit noise was more considered and could benefit from error correction techniques. Quantum volume (QV) has turned into an important parameter apart from the qubit number since quantum computer developers considered that it was also important to have well-calibrated controlled noise qubits rather than ti have a huge number of noisy qubits. QV measures the performance of gate-based quantum computers [21]. Therefore, to achieve supremacy, it is necessary to have a higher number of qubits that are well-calibrated. Another crucial metric to measure quantum device performances were introduced by IBM recently. CLOPS (circuit layer operations per second) corresponds to the number of quantum circuits a quantum processing unit (QPU) can execute per unit of time. In this regard, and according to the available systems (ibm_perth, ibm_lagos, ibm_jakarta with seven qubits; ibm_manila, ibm_bogota, ibm_quito, ibm_belem, and ibm_lima with five qubits), a representation of CLOPS vs time is presented in Figure 7.

The trend is that a system with higher CLOPS solves the problem faster; however, this statement is not always true. This is due to the internal architecture of each quantum computer and how the connections between the qubits are done. A low CLOPS QC can be fast for a determined problem but slow for another one. For each problem, the performance can vary since CLOPS is determined using a general purpose test created by IBM. As a result, for this particular routing problem, the Belem and Quito architectures and qubit connections seem to perform better than the Lima or Manilla ones, even though their CLOPS systems in the IBM tests were higher.

The results correspond to the single objective problem. The same examination could be applied to different problems, depending on the qubits available. The outcomes reveal that CLOPS also play a role, as higher CLOPS systems tend to solve the problem faster on average, even though more qubits can affect the system coherence. With 7 qubits and 2.9K as CLOPS, the time decreases as Figure 6 reveals.

Moreover, simulations performed in [22] conclude that QAOA will achieve a quantum speedup with hundreds of qubits since classic computers have begun to struggle with the size of the problem. [23] also discusses an approach to supremacy for optimization problems making use of QAOA. This algorithm is designed to run purely on a gate model quantum computer, and it is hard to simulate by any simulator when a high number of qubits are involved. Although there exist classical natural algorithms that have better success chances, QAOA has a performance guarantee. As a result, it is argued if QAOA can exhibit any form of “quantum supremacy”, with the conclusion being positive due to the theoretical complexity assumptions. The paper postulates QAOA as an excellent candidate to run on near-term quantum computers, not only because of the potential use in optimization for some particular problems but for the very probable supremacy demonstration once the hardware has small hundreds of intermediate-noise qubits. Since the device availability only allows tests on IBM quantum systems with few qubits and good noise levels, theoretical estimations need to be performed.

Figure 8 shows supremacy expectations on node routing problems considering IBM quantum computers, which are the ones used in this research.

The red bar indicates the executions done in this paper on open-for-research IBM QC, where the time taken to find the solution is relatively low but still much higher compared to the classic solution. The yellow space indicates where IBM is working now, testing systems up to 65 qubits (Hummingbird quantum processor) and acceptable levels of noise to allow accurate calculus. The green space indicates the estimations of supremacy, making use of QAOA to solve the routing problem. The plot in Figure 8 providing estimates was calculated based on the complexity analysis performed on [23]. The mathematical and quantum physics principles behind the QAOA algorithm make such low complexities possible for the application to this kind of problem, where the main handicap is the difficulty in simulating it on a classical computer. However, the tests performed so far on IBM hardware proved the small complexity of the problem; to perform the supremacy estimations, the trend was extrapolated to bigger problems using regression techniques. This was compared with a classical implementation of the problem on a Python general purpose linear programming kit (GPLK) to check the computational cost of this implementation compared with the quantum solution. Afterward, supremacy expectations were achieved by comparing costs and positioning the quantum implementation in the IBM real hardware road map. As a result, the research determines that a few hundred qubits (well-calibrated) are enough to reach supremacy for this kind of problem, where classic algorithms really struggle to reach optimal solutions. This is expected to be reached when an Eagle 127 qubits processor achieves stability in a 2–3 year vista. Some factors that may move the exact supremacy point are the CLOPS and the volume achieved in the Eagle 127 qubits processor. Higher CLOPS will decrease the time used to calculate the solution, and a higher quantum volume will increase the performance of the algorithm by obtaining better solutions with fewer iterations.

## 4. Conclusions

Optimization problems have always been challenging for computers due to the inherent difficulty involved. Quantum computers have already shown their potential in many application fields, with optimization tools being an advanced area. The next generation of computers will be able to use other methods to solve hard optimization problems. Quantum computers have shown their potential for these purposes, making use of new algorithms designed to take profit from quantum properties. QAOA is an example of those algorithms and has shown strong potential since its release. This results in fast, powerful algorithms for quantum computers (that nowadays lack the hardware to reach their true potential). In this case, QAOA has been used to solve some routing problems applicable to the next generation of wireless communication systems. Since 6G will include the massive interconnections between multiple nodes, optimal routing will have a large importance in resource optimization. This paper has shown that it is already feasible to solve routing problems with QAOA. The first example introduced the entire procedure followed by solving the single-target routing problem; the second case involved a higher number of nodes and multi-objectives to test QAOA and provide conclusions. Moreover, as better devices are available, fewer iterations of the algorithm will be needed. Routing problems have additional difficulty since integer programming is harder to solve than linear programming, but QAOA still managed to find optimal routes for the problems proposed. These examples were small demos of what a bigger problem would look like, but the actual quantum computers can only solve limited-size problems. The scalability of the problems solved using QAOA rely on the new quantum hardware available. A higher number of qubits will allow bigger problem resolutions, and soon a quantum computer could reach the size where it outperforms classical computations in terms of solving time. The higher quantum volume will enhance accuracy and, as a result, will decrease the number of iterations of QAOA needed to solve the problem. Higher CLOPS will speed up any process in the quantum computer and, as a result, will enable real-time calculations. The research topic is still open. In this paper, the results were obtained from the lexicographic method to find the Pareto point in the multi-objective problem. Another method, such as goal programming, could also be used in this framework. The number of qubits required for a 6-node example was 50 for this method. It could not be run on a quantum computer, but its implementation will be under consideration for further study.

## Figures and Tables

**Figure 1 sensors-22-07570-f001:**
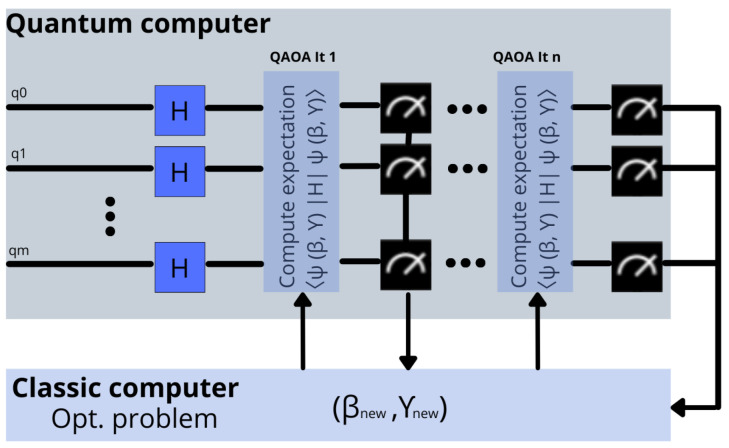
Graphical representation of the operation principle of a QAOA scheme.

**Figure 2 sensors-22-07570-f002:**
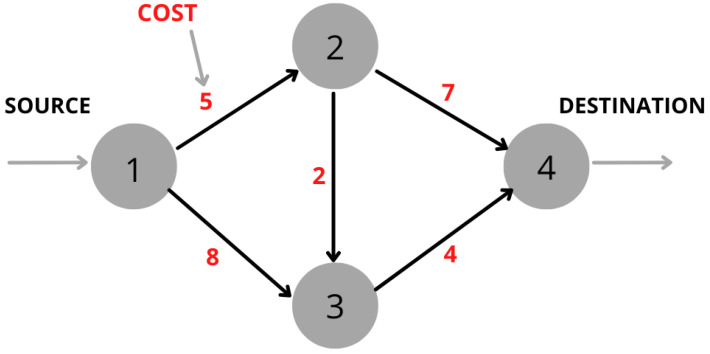
Scheme of an example network with four nodes.

**Figure 3 sensors-22-07570-f003:**
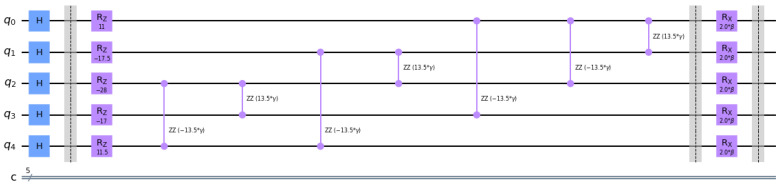
Representation of the single-objective, four-node example QAOA circuit.

**Figure 4 sensors-22-07570-f004:**
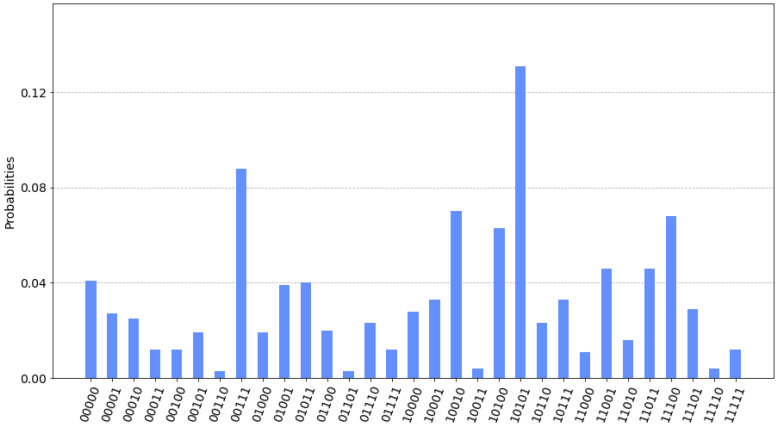
Probability results for the example network.

**Figure 5 sensors-22-07570-f005:**
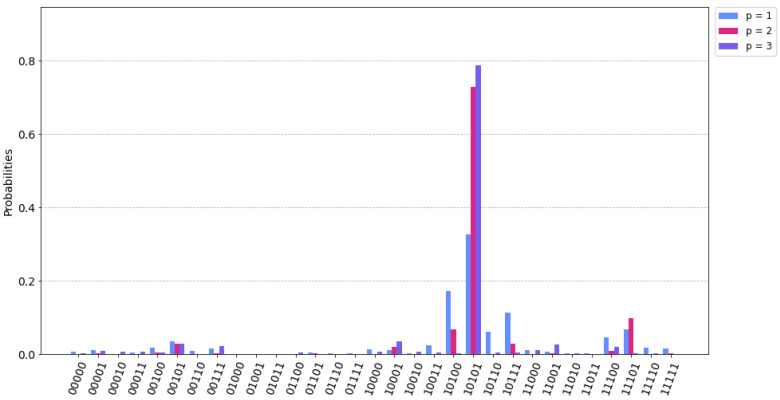
QAOA performance comparison according to p values.

**Figure 6 sensors-22-07570-f006:**
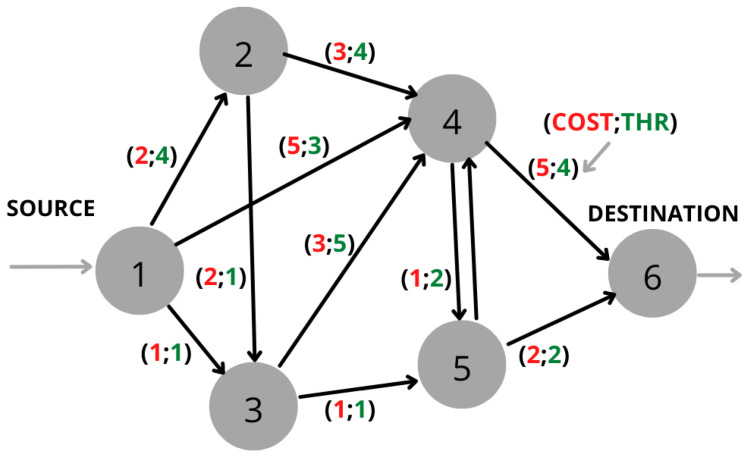
Scheme of an example network with six nodes.

**Figure 7 sensors-22-07570-f007:**
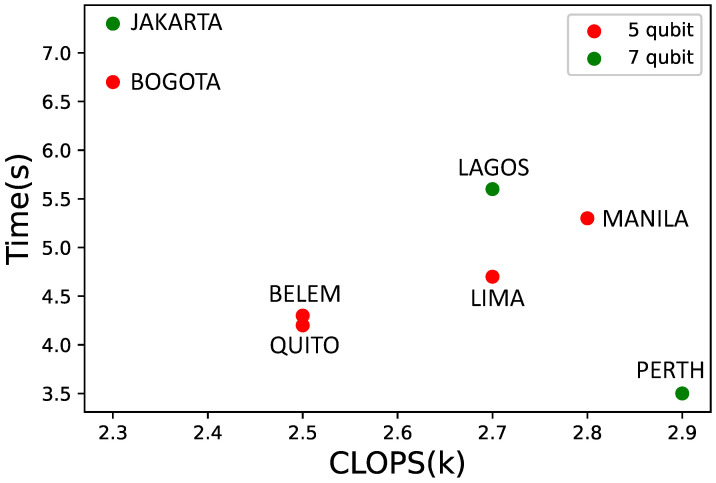
Time for solving the four-node problem depending on QC CLOPS.

**Figure 8 sensors-22-07570-f008:**
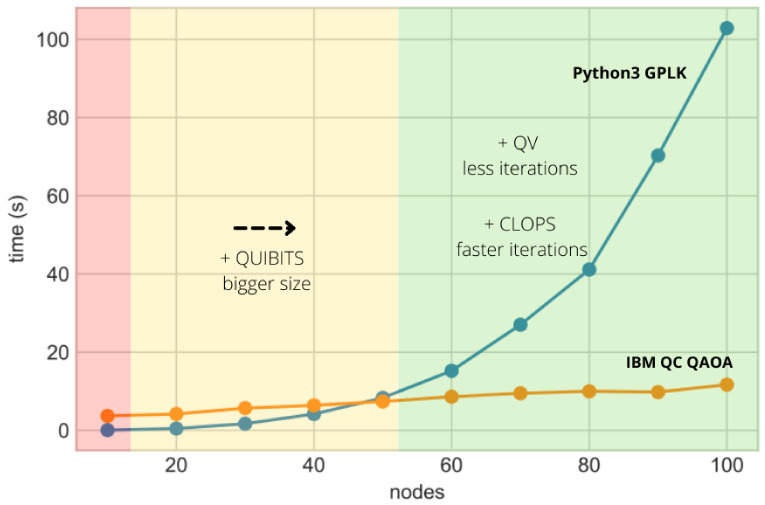
QAOA Supremacy estimations for network routing problems.

**Table 1 sensors-22-07570-t001:** Results obtained for the six-node lexicon optimization.

Objectives	Min Cost	Max Thr st Cost	Min Hops st Cost+Thr
Cost	4	7	4
Throughput	4	10	4
Hops	3	4	3

## Data Availability

Not applicable.

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
