# Peer review of "Multi-Objective Routing Optimization for 6G Communication Networks Using a Quantum Approximate Optimization Algorithm"

_sensors, 2022, doi:10.3390/s22197570_

Round 1

Reviewer 1 Report

In this work entitled “Multi-Objective Routing Optimization for 6G Communication Networks Using Quantum Approximate Optimization Algorithm”, in order to solve single and multi-objective network routing problems, the authors focused on the routing problems in a 6G network scenario using QAOA, and presented the solution to some routing problems applicable in the next generation of wireless communication systems. They also performed the algorithm on the 7-qubit IBM quantum computer.

 In my opinion, the current manuscript contains quite an important and interesting topic. They performed the QUBO algorithm on the IBM processor and the related results are shown. This shows the advantages of quantum computing on the acceleration using existing quantum algorithms. Also, the manuscript is well organized and clearly written. I suggest that it can be accepted after the authors considering the following issues:

In the first part, the authors introduced the background of NISQ, I think it is quite simple and there are several important and related references are missing.

The references should be cited in sequence, and a typo of the authors name in ref.[12].

Table 1 is not clearly described. I suggest the authors to give a detailed description.

In fig.6, there is not clearly correspondence of the IBM system with the red and green points. Could you change the expression and denote the related systems in it?

What does Python GPLK and the related points in figure 7 mean?

Author Response

Find the detailed response in the attachment. 

Reviewer 2 Report

In this work, the authors study the quantum approximate optimization algorithm (QAOA) for routing problems. Recently, research on the application of small-scale quantum computers has attracted more attention due to its potential e.g. quantum supremacy. When 6G is realized, the routing problem in wireless multi-hop networks needs to be solved, even if only approximately. The authors investigate the performance of QAOA for the routing problem, which may have advantages over classical computers.

Although their method is limited to routing problems, the present manuscript will provide meaningful motivation to apply QAOA to a variety of other problems. This manuscript is clearly written, with nice readability, and the technical content is understandable. Thus, I recommend the publication of this manuscript after considering optional (minor) changes.

1. In line 276, the authors comment that higher CLOPS systems tend to solve the problem faster on average. However, that argument may not be valid in the case of 5 qubits. If possible, I suggest that the authors should describe why higher CLOPS increase time in the range from 2.5 to 2.8 k.

2. In Fig.7, the authors show the results of QAOA Supremacy estimations. The authors comment that the data for lager qubit systems is estimated from that demonstrated by few qubits systems, while I did not understand well how the estimation is performed. It would be helpful if the authors could describe the process for the estimation in more detail.

Author Response

Find the detailed replies in the attachment.

Reviewer 3 Report

In this paper, the authors addressed routing problem using 6G and quantum computing. This paper presentation is good and the contributions seems to be novel. Since the authors provided illustrative examples, and they help to understand the solution clearly. Beside, there are few comments to be addressed before consider this paper for publication.

1. The remaining section details in the last paragraph of the introduction to be written as separate paragraph

2. The contributions of the paper are summarized using a list.

3. The literature of the paper is missing. It is recommended to provide the literature of the paper by considering most recent papers. It is also recommended to summarize all the limitations of the existing routing process and how the proposed work is better over them.

4. Some of the mathematical formulae available in existing and published papers. It is recommended to provide the citations for them.

5. There are several metrics in the literature to judge the performance of the algorithm, whereas the authors did very few in the paper. More results under different scenarios are evaluated.

6. The source code of the simulations are embedded in the paper or cited in the paper.

7. The reasons for achieving the superior performance of the proposed work over the existing ones to be listed in the paper.

8. The conclusion can be written as single paragraph. The remaining discussion can be written as a separate subsection under the results section.

Author Response

(The authors gave the same response as above.)

Round 2

Reviewer 3 Report

The authors addressed all the recommended comments and the current version of the paper is well improved over the previous version. So, I recommend this paper for publication in this journal. Congratulations to the authors.